# Hot Topics on COVID-19 and Its Possible Association with Guillain-Barré Syndrome

Anelia Dietmann [1,*,†], Paolo Ripellino [2,†], Andrea M. Humm [3], Thomas Hundsberger [4], Bettina Schreiner [5], Marie Théaudin [6] and Olivier Scheidegger [1]

1  Department of Neurology, Institute of Diagnostic and Interventional Neuroradiology, Inselspital, University Hospital and University of Bern, 3010 Bern, Switzerland; olivier.scheidegger@insel.ch
2  Department of Neurology, Neurocenter of Southern Switzerland EOC, 6900 Lugano, Switzerland; paolo.ripellino@eoc.ch
3  Unit of Neurology, Department of Internal Medicine, HFR Fribourg-Hôpital Cantonal, 1708 Fribourg, Switzerland; andrea.humm@h-fr.ch
4  Department of Neurology, Cantonal Hospital St. Gallen, 9007 St. Gallen, Switzerland; thomas.hundsberger@kssg.ch
5  Department of Neurology, University Hospital Zurich, 8091 Zurich, Switzerland; bettina.schreiner@usz.ch
6  Nerve-Muscle Unit, Department of Clinical Neurosciences, Lausanne University Hospital CHUV and University of Lausanne, 1011 Lausanne, Switzerland; marie.theaudin@chuv.ch
*  Correspondence: anelia.dietmann@insel.ch; Tel.: +413-1664-0613
†  These authors contributed equally to this work.

**Abstract:** As the COVID-19 pandemic progresses, reports of neurological manifestations are increasing. However, despite a high number of case reports and case series on COVID-19 and Guillain-Barré-Syndrome (GBS), a causal association is still highly debated, due to the lack of case-control studies. In this opinion paper, we focus on a few clinically relevant questions regarding the possible link between GBS and SARS-CoV-2 infection or vaccination based on our personal clinical experience and literature review.

**Keywords:** Guillain-Barre syndrome; COVID-19; immune-mediated neuropathy

## 1. Does SARS-CoV-2 Infection Cause GBS?

Despite a high number of case reports and case series on COVID-19 and Guillain-Barré-Syndrome (GBS), a causal association is still highly debated, due to lack of case-control studies [1–4]. GBS is a monophasic autoimmune disease characterized initially by rapidly progressive, mostly symmetrical weakness of the extremities. It is in up to two-thirds of patients preceded by an infection [5]. Many infectious agents have been associated with GBS, such as *Camplyobacter jejuni*, Epstein–Barr virus, influenza virus, and Zika virus, amongst others [6–8]. A potential association between GBS and COVID-19 raised physicians' attention during the first wave of the COVID-19 pandemic, both in China [9] and in Europe [10–13]. In 2020, GBS was reported in 0.1–0.4% of patients hospitalized for COVID-19 [14] and represents, together with its variant Miller-Fisher syndrome, about 20% of the neurological case reports [15]. However, we suggest maintaining a critical appraisal when approaching these data, especially considering the frequency of other causes of muscular weakness in COVID-19 patients, such as critical illness neuropathy/myopathy [16]. At least in the first months of 2020, the diagnostic certainty, scored according to Brighton collaboration criteria [17], was lacking for many of the reported GBS cases because lumbar puncture or a rigorous neurophysiological examination were not performed due to the emergency setting caused by the pandemic. The same applies to the diagnostic confirmation of the "preceding SARS-CoV-2 infection," often defined only based on clinical signs without microbiological verification due to the shortage of tests during March to April 2020. To establish the degree of certainty of the association, Ellul et al. [18] proposed to consider a

"probable association" between GBS and SARS-CoV-2 infection in case of: (1) neurological disease onset within 6 weeks of acute infection; (2) either SARS-CoV-2 RNA detected in any sample (e.g., positive RT-PCR from nasopharyngeal swab, cerebrospinal fluid, or serum) or antibody evidence of acute SARS-CoV-2 infection (IgM, IgG seroconversion, or an increase of four times in antibody titers in paired acute and convalescent serum samples); and (3) no evidence of other commonly associated causes (*Campylobacter jejuni*, *Mycoplasma pneumoniae*, Cytomegalovirus, Epstein–Barr virus, hepatitis E virus, Zika virus, or HIV infections). Even with evidence of other commonly associated causes, the association should be considered as "possible" if conditions (1) and (2) apply.

An increase in GBS incidence in COVID-19 patients was found in studies from Italy, Spain, and France during the first wave peak (March–April 2020) relative to previous years. Gigli et al. [19] observed increased numbers of GBS cases in a northeastern region of Italy from March to April 2020, with a mean of 3.5 cases/months compared to 0.67 cases/months in the previous years. However, only 3 out of 8 GBS cases with possible association to COVID-19 had clinical signs/symptoms of COVID-19 infection; none had positive swab tests or SARS-CoV-2 PCR, and only one had a positive SARS-CoV-2 serology.

In the same period (i.e., March–April 2020), Filosto et al. [13] found an overall increase in GBS incidence compared with 2019 in a retrospective analysis of 34 GBS cases (of whom 30 were COVID+) diagnosed in referral hospitals from Northern Italy. The study calculated an estimated incidence of GBS of 47.9 cases per 100,000 SARS-CoV-2 infections. Again, in the same time frame (i.e., March–April 2020), a retrospective, case-control, multicentric study [20] from Spain analyzed the incidence of GBS in a total number of over 1.4 million patients presenting to one of 61 Spanish emergency departments, comparing COVID+ and COVID- patients. Compared to 0.15‰ of COVID-19 patients ($n = 11/71{,}904$), only 0.02‰ of non-COVID-19 patients ($n = 33/1{,}358{,}134$) had GBS. Therefore, COVID-19 infected patients had a significantly higher relative frequency of GBS with an odds ratio of 6.3. However, it has to be taken into account that these studies had small patient numbers with a short observation period and had several potential confounders due to the retrospective evaluation.

In fact, two cohort studies found no significant increase in number of GBS patients during the first wave of the pandemic compared to previous months. In the first study, Keddie et al. [21] analyzed the epidemiology of GBS cases reported to the United Kingdom National Immunoglobulin Database during the pandemic relative to previous years (2016–2019). They report a reduction of GBS cases during the COVID pandemic compared to previous years. Keddie and colleagues reasoned that the influence of lockdown measures reducing the transmission of pathogens such as *Campylobacter jejuni* and other respiratory viruses might have influenced the reduced number of reported GBS cases during the 1st wave of the COVID-19 pandemic [21]. In the same study, a subset of 47 patients diagnosed with GBS between March and May 2020 and seen by members of the British Peripheral Nerve Society is described. Among the 47 cases, 25 (53%) had a confirmed/probable SARS-CoV-2 infection [21]. From these observations, the authors concluded that a causative relationship between SARS-CoV-2 and GBS was unlikely and that COVID associated GBS does not present with a specific clinical phenotype.

The second study [22] analyzed the data collected between 30 January 2020 and 30 May 2020 within the prospective International GBS Outcome Study (IGOS). Overall, 49 GBS patients were included, of whom 8 (16%) had a confirmed and 3 (6%) a probable SARS-CoV-2 infection. Nine of these 11 patients had no serological evidence of other recent preceding infections associated with GBS, whereas two had serological evidence of a recent *Campylobacter jejuni* infection. Patients with a confirmed or probable SARS-CoV-2 infection frequently had a sensorimotor variant 8/11 (73%) and facial palsy 7/11 (64%). All patients who underwent electrophysiological examination had a demyelinating subtype. The median time from the onset of infection to neurological symptoms was 16 days. Given the fact that, in this study, no increase in inclusion rate was found, it seems that the risk of developing GBS following SARS-CoV-2 infection is small, considerably lower when

compared with, for example, *C. jejuni* or Zika virus. Finally, the colleagues from Singapore also reported in a letter a decrease in the number of GBS cases during the COVID-19 pandemic based on a national registry [23].

To our knowledge, detection of other recent preceding infections associated with GBS in cases of COVID-19 associated GBS have only rarely been described so far [11,13,20–22,24].

To conclude, although a relationship between GBS and SARS-CoV-2 infection cannot be ruled out based on the currently available literature, a strong association seems unlikely.

## 2. Clinical Features of GBS Related to COVID

Overall, the onset of COVID related GBS seems to emerge within 7–14 days after COVID symptoms appear (cough, fever, anosmia) [24], but it is possible that even COVID+ but asymptomatic patients develop neurological symptoms. The possible pathogenesis of COVID related GBS is a matter of discussion (para-infectious vs. post-infectious), but is not the focus of this paper [25]; we only acknowledge here that SARS-CoV-2 RNA has—to our knowledge—only been described in one case of CSF in any case of reported COVID related GBS [20].

The majority of reported COVID related GBS patients suffer from a sensorimotor syndrome, with prominent albumin-cytological dissociation, and are classified electrophysiologically as acute inflammatory demyelinating polyneuropathy (AIDP), the classical demyelinating variant [21,22,24,26,27]. Clinical course and disease severity of COVID related GBS are comparable to classic GBS [28].

However, based on our literature review [27,29] and personal experience [22], when GBS is occurring together with COVID, it is quite common to observe a facial nerve palsy, often bilateral, and to have an early and severe autonomic dysfunction.

Uncini et al. [26] compared neurophysiological features of 24 COVID related GBS vs. 48 control cases of demyelinating GBS. The most common pattern encountered in COVID related GBS is distal compound muscle action potential (dCMAP) duration increase and the absence of F waves, but persistence of normal distal motor latencies (DML).

A systematic review on treatment options for COVID-19 related GBS evaluated 63 studies (overlapping sources with the systematic review from Abu-Rumeileh et al. [30]) and included 86 patients with an analysis of outcome data in 76 cases [31]. A mortality rate of 3.5% was found in patients with COVID-19 related GBS, which is higher than in COVID-19 symptomatic infection cases (2.2%) [31] as well as in non-COVID-19 related GBS cases (2.8%) [32]. Similar to non-COVID-19 related GBS cases [32], risk factors for poor outcome were older age, quadriplegia, and respiratory failure [31]. Regarding treatment options for COVID-19, systemic steroids did not affect mortality or admission to intensive care unit in patients with COVID-19 and GBS. The use of intravenous immunoglobulins (IVIG) was reported in 87% of cases, without clear evidence for improved outcome or reduced mortality [31].

To summarize, reported COVID-19 related GBS patients mainly have a classical demyelinating AIDP phenotype that starts, on average, within 1–2 weeks after COVID and even in patients with mild/recovering COVID symptoms. Facial nerve palsy, often bilateral, and autonomic dysfunction may occur more frequently. Thus far, no specific treatment guidelines for COVID-19 related GBS cases have been established, but as the assumed pathogenesis and clinical phenotypes that are similar to classical GBS, IVIG, or PEX should be considered depending on the clinical severity of neurological deficits and the availability of treatment modality, respectively.

## 3. Is SARS-CoV-2 Vaccination a Risk Factor for the Development of GBS?

Theoretically, the vaccination could stimulate the immune system and trigger neurological adverse events as it happened in 1976 with the influenza vaccine in the United States, with an increase of 1 attributable case of GBS per 100,000 vaccinated persons [33]. Hence, similar concerns on the safety of COVID-19 vaccines might arise in the general population and among patients already suffering from neurological diseases [34].

Furthermore, recent case reports claim an association of Bell´s palsy with mRNA SARS-CoV-2 vaccination and development of a neuromyelitis spectrum disorder (NMSD) in a patient with stable multiple sclerosis after vector-based SARS-CoV-2 vaccination, respectively [35,36].

There are currently some case reports of GBS after SARS-CoV-2 vaccination [37–39]. However, temporal association does not mean causation [40] and caution in its interpretation is advised [41,42], as large studies are lacking.

When considering the risk of GBS after COVID vaccination, the most important consideration to be made is that the benefits of the vaccination largely outweigh the risks, because the vaccination is the most effective public health intervention we have to fight the pandemic and its global medical and economic burden. The individual risk to suffer from GBS is very small, whereas the advantage of the vaccination is large both for the individual and for the community.

For these reasons, experts on inflammatory neuropathies from different geographical areas strongly encourage vaccination against COVID-19 [40,41,43,44].

On the other hand, it is important to remain vigilant and transparent [42,45] and continue to collect timely and precise pharmacovigilance data. Potentially, neurological and other side effects may be underreported or delayed due to various reasons (timely reporting procedure, additional working load, neglect of a potential association, administrative delay due to inefficient reporting procedures). Hence, it is noteworthy that in Vigibase, a large global database on vaccine-related side effects, GBS represents 0.1% (1/1000) of all the COVID-19 vaccine-related safety reports [46].

Another important question is whether the risk of GBS depends on the vaccine type. The US Food and Drug Administration (FDA) has raised some concern regarding ChAdOx1-S (Astra Zeneca) and Ad26.COV2-S (Janssen/Johnson & Johnson) vaccination [47], and further studies [48,49] confirmed the existence of a small association.

In a multi-institutional study in Taiwan [49], 1 GBS case was identified after a first dose of ChAdOx1-S (Astra Zeneca) vaccine out of over 18,000 healthcare workers who received the first dose of the same vaccine. The same group performed a systematic review of 17 publications on 38 cases of GBS related to COVID-19 vaccination, and 25 were reported thereof after ChAdOx1-S, 12 after Pfizer BioNTech, 1 after Johnson & Johnson, and 1 after CoronaVac [49]. In this review, the average time from vaccination to symptom onset was 11 days and the patients presented with a classical GBS clinical phenotype in most cases [49].

A potential small but significant association of GBS following the Ad26.COV2-S (Janssen, Johnson & Johnson) vaccine has been found within the US Vaccine Adverse Event Reporting System (VAERS) with 130 cases of presumptive GBS from February to July 2021 after receiving the Ad26.COV2-S vaccine [48]. With approximately 13,209,858 doses of vaccine administered to adults in the US, the estimated crude reporting rate was 1 case of GBS per 100,000 doses administered. This study found an absolute rate increase of 6.36 per 100,000 person-years [48]. Together with its increased risk for thrombosis with thrombocytopenia syndrome, the Advisory Committee on Immunization Practices of the United States (CDC) recommends mRNA COVID-19 vaccines over the Janssen COVID-19 vaccine for the full vaccination schema or booster doses [50].

In an analysis of 3,890,250 recipients of the BNT162b2 mRNA vaccine, only seven cases of GBS were detected after the first dose within 30 days (observed incidence 0.18/100,000) and no case was reported after the second dose [51]. Recently, in a retrospective analysis of 700 patients from Israel who had previously been diagnosed with GBS (between 2000 and 2020) and who received a mRNA COVID-19 vaccine (BNT162b2, Pfizer BioNTech, only one patient presented with a GBS relapse after the second dose [52].

Finally, it is interesting to note that facial diplegia [37,38,53–55] is relatively often encountered in GBS related to COVID vaccination, and is also observed in COVID-related GBS [56].

To summarize, GBS (also in the form of facial diplegia) may rarely occur within approximately 6 weeks of COVID-19 vaccination. Overall, mRNA COVID-19 vaccines are recommendable over vector-based vaccines. Further immunological studies analyzing the possible pathophysiology underlying these events (in the context of infection/vaccination) are currently ongoing. Since the consequences of COVID-19 far outweigh the risks of SARS-CoV-2 vaccination, current guidelines [41,43] recommend most of our neuromuscular patients to get vaccinated.

**Author Contributions:** Conceptualization, literature research and selection of literature, writing—original draft preparation A.D. and P.R. Writing—review and editing A.M.H., T.H., B.S., M.T. and O.S. All authors have read and agreed to the published version of the manuscript.

**Funding:** This research received no external funding. A.D. is personally funded by a protected research time grant from the University Bern. P.R. acknowledges "GBS/CIDP Foundation", "Baasch-Medicus Stiftung", "Foundation for the Advancement of Neurology" and "GBS initiative Schweiz" for their support to research in the field of GBS in Switzerland.

**Institutional Review Board Statement:** Not applicable.

**Informed Consent Statement:** Not applicable.

**Data Availability Statement:** Not applicable.

**Conflicts of Interest:** The authors declare no conflict of interest.

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
