# Peer review of "Hot Topics on COVID-19 and Its Possible Association with Guillain-Barré Syndrome"

_ctn, doi:10.3390/ctn6010007_

Round 1
Reviewer 1 Report
The opinion paper aims to clarify the existence or not, of a causal association between COVID-19 and Guillain-Barré-Syndrome (GBS). This paper, despite the lack of case-control studies focus on few clinically relevant questions regarding the possible link between GBS and SARS-CoV-2 infection or vaccination based on their personal clinical experience and literature review.
The opinion paper is clear, relevant for the field and presented in a well-structured manner. The cited references are current and it doesn’t include an abnormal number of self-citations. The scientific sound and the experimental design of the opinion paper are appropriate to present the theme. The structure is the appropriate and the conclusions and statements drawn coherent and supported by the listed citations The literature review is clear, comprehensive and of relevance to the field. Also, there is a gap in knowledge identified at this field.
The suggestion is that those lines :
|
GBS is a monophasic autoimmune disease characterized initially by rapidly progres- |
38 |
|
sive, mostly symmetrical weakness of the extremities. It is in up to two-thirds of patients |
39 |
|
preceded by an infection[5] |
Should strengthen, with few details about the infections that are associated with GBS.
Author Response
Dear Reviewer,
we thank you for your work and appreciate your comments and we added important information on GBS and infectious agetns accordinly. The manuscript now includs a sentence on other infectious agents associated with GBS including references.
Kind regards,
Anelia Dietmann and Olivier Scheidegger
Reviewer 2 Report
The author discusses in detail the relationship between GBS and COVID-19 that has been discovered so far, enabling us to learn about the connection between the current COVID-19 epidemic and the mechanisms of neurological disease.Here are some of my suggestions.
1. In the first part of the article, the author mentioned that novel coronavirus infection may cause the occurrence of GBS. Are there any other possible sources of infection in these cases reviewed in our previous clinical experience?
2. The author mentioned that according to his own clinical experience, which was not reflected in the article, does the author have any relevant clinical data?
Author Response
Dear Reviewer,
we thank you for your work and your comments. The first suggestion points to an important issue - other infections that may have caused or contributed to the development of GBS at the same time as the COVID-19 infection. Interestingly, in the Paper from Fragiel et al. Ann Neurol. 2021 no COVID-GBS patient had another positive result from serology or conventional cluturs indicating another infectous agent, however in the Non-COVID-GBS group, there was one C. jejuni infection, 1 B. burgdorferi infection and 1 CMV infection. Other studies such as Filosto et al. JNNP 2021 or Foresti et al. EJN 2021 do not give any details on testing for other infectious agents. Abu-Rumeileh et al. JoN 2021 summarizes all Case Reports on COVID-19 GBS from 01/2020 until 07/2020 and lists all given information on testing for other infectious agents and found no case report on a confirmed co-infection. In the study from Luijten et al. Brain 2021, nine out of 11 patients with COVID-19 associated GBS had no serological evidence of other recent preceding infections associated with GBS, whereas two had serological evidence of a recent Campylobacter jejuni infection. We added this information to the manuscript.
We refer to our own patients treated for GBS associated with GBS that are all included in the IGOS study database (Luijten et al. Brain 2021).
We hope to have answered your comments satisfactorily and thank you again for your work.
Kind regards
Anelia Dietmann and Olivier Scheidegger